# Spatial coherence of light inside three-dimensional media

Marco Leonetti [1,2✉], Lorenzo Pattelli [3,4], Simone De Panfilis [1], Diederik S. Wiersma [3,4,5] & Giancarlo Ruocco[1,6]

Speckle is maybe the most fundamental interference effect of light in disordered media, giving rise to fascinating physical phenomena and cutting edge applications. While speckle formed outside a sample is easily measured and analysed, true bulk speckle, as formed inside random media, is difficult to investigate directly due to the obvious issue of physical access. Furthermore, its proper theoretical description poses enormous challenges. Here we report on the first direct measurements of spatially resolved intensity correlations of light inside a disordered medium, using embedded DNA strings decorated with emitters separated by a controlled nanometric distance. Our method provides in situ access to fundamental properties of bulk speckles as their size and polarization degrees of freedom, both of which are found to deviate significantly from theoretical predictions. The deviations are explained, by comparison with rigorous numerical calculations, in terms of correlations among polarization components and non-universal near-field contributions at the nanoscale.

[1] Center for Life Nano- & Neuro-Science, Fondazione Istituto Italiano di Tecnologia (IIT), Roma, Italy. [2] Soft and Living Matter Laboratory, Institute of Nanotechnology, Rome, Italy. [3] Istituto Nazionale di Ricerca Metrologica (INRiM), Torino, Italy. [4] European Laboratory for Non-linear Spectroscopy (LENS), Sesto Fiorentino, Italy. [5] Dipartimento di Fisica, Università di Firenze, Sesto Fiorentino (FI), Italy. [6] Dipartimento di Fisica, Università "La Sapienza", Roma, Italy. ✉email: marco.leonetti@iit.it

When subject to multiple scattering, propagating electromagnetic waves will, in general, accumulate random and uncorrelated phase delays resulting in a fluctuating profile known as speckle pattern. The existence of these granular structures became apparent since the invention of laser[1,2] and yet up to these days we still keep learning about their fundamental properties[3–5] and the range of applications they enable, spanning from imaging[6–10] to spectroscopy[11] and cryptography[12], to name a few.

The key features of speckle patterns are the length scales over which they fluctuate, which can be measured looking at the decay of the intensity spatial correlation function $C_I(\mathbf{r}_1, \mathbf{r}_2) = \langle I(\mathbf{r}_1)I(\mathbf{r}_2)\rangle$. While the advent of near-field microscopy has pushed the limits of such measurements down to the immediate vicinity of the outer surface of a scattering material[13–16], most information about the nature of spatial intensity fluctuations inside a three-dimensional (3D) medium remains precluded due to the invasive nature of conventional near-field detection techniques[17,18]. As a matter of fact, even nonspatially resolved information available to date is limited to lifetime statistics studies[19,20].

In addition to the experimental difficulties, also the theoretical description of light propagation inside 3D nanostructured material is a challenging task, requiring to address a broad range of length scales (from subwavelength resonances to the scaling of transport properties to increasing system size), coupling of vector-wave components in three dimensions, nonuniversal, and possibly even nonlocal effects[14,21,22]. As a result, certain optical properties of complex media remain particularly hard to investigate even numerically, which explains why we have only recently started to unveil several unexpected aspects of light propagation and transmission inside disordered media[23–27].

Gaining access to the distribution of light inside complex photonic media is important for a series of reasons. Connecting the spatial and polarimetric properties of bulk light fields to the structural information of a sample will deepen our understanding of optical transport phenomena, effective-medium theories, intensity, and polarization correlations. More importantly, it will allow us to relate the optical properties that we observe from the outside to what is happening inside a photonic medium, which can help the design of new materials and devices with tailored photonic functionalities.

In this work, we present an experimental study, supported by numerical calculations, on two-point intensity correlations of the optical fields inside bulk 3D random media. We use a new technique (also introduced in this paper) that allows to measure such correlations at a nm-controlled distance, based on embedded emitters separated by DNA strings of calibrated length. The experimental results for both scattering and non-scattering media are compared to full-wave 3D numerical calculations which allows to unveil several unexpected phenomena. For a simple scattering medium made of dielectric nanoparticles in an aqueous environment, we find that the size of the speckle that is formed inside the random sample is almost two times smaller than that expected from existing theories—a result which we can attribute to a combination of polarization and near-field effects. We also reveal information about the local direction of the polarization in the proximity of subwavelength scattering elements, and find nonuniversal correlations between its components. The length scale of the intensity correlations that we find is well below the diffraction limit: the technique that we introduce here is in fact not limited by diffraction and can, in principle, be applied also to other, non-disordered, 3D photonic structures. This could prove particularly useful given the increasing evidence (so far limited on lower dimensional systems) that even extreme subwavelength deviation from an intended design can give rise to striking and unpredictable deviations from the expected photonic behavior[28–30]. Overall,

the measurements on which we report here provide a new type of experimental test bench for the validity of theoretical models in the mesoscopic scattering regime and their dependence on the microscopic details of the scattering medium.

## Results

In a general description of a propagating light field, the probability density function (p.d.f.) for the normalized intensity $I = I_{\text{raw}}/\langle I_{\text{raw}}\rangle$ (with $I_{\text{raw}}$ the as-measured intensity) of a fully developed speckle pattern is described by the following expression[31]:

$$p(I) = \frac{N^N}{\Gamma(N)} I^{N-1} e^{-NI}, \tag{1}$$

where $\Gamma$ is the Euler's Gamma function and $N$ the number of "non interfering" contributions as given by e.g., different lasers, different wavelengths, or polarizations. As can be seen from Eq. (1), even though the p.d.f. does not carry information about the spatial distribution of the light intensity, it is extremely sensitive to the number $N$ of statistically independent speckle contributions.

The basic idea of our approach relies on measuring the $p(I_{\text{tot}}) = p(I(\mathbf{r}_1) + I(\mathbf{r}_2))$ resulting from intensities at two different locations separated by a known distance $D = |\mathbf{r}_2 - \mathbf{r}_1|$ inside the same speckle pattern. Such p.d.f. is built up by collecting intensity measurements at fixed pairs of locations for different and uncorrelated illumination realizations[32] (each labeled with a discrete index $t$). This is achieved experimentally using a digital micromirror device (DMD) in the super-pixel method configuration (see Methods and Supplementary Fig. 1), which allows to rapidly scramble the phases of the illumination wave at each measurement. Given a certain speckle grain size $S$, we therefore expect to observe a gradual transition between $N = 1$ for $D \ll S$ to $N = 2$ when $D \gg S$. For large $D$, $p(I_{\text{tot}})$ will result from the sum of two uncorrelated quantities. On the other hand, at $D \ll S$ we will have $I(\mathbf{r}_1) \simeq I(\mathbf{r}_2)$ and thus $p(I_{\text{tot}}) \simeq p(2I(\mathbf{r}_1)) = p(I(\mathbf{r}_1))$, leading to $N = 1$. This allows to experimentally determine $S$ by measuring intensity correlations between separated emitters (ICSE) and fitting Eq. (1) to the resulting p.d.f. using $N$ as the only free parameter.

**Nanorulers in a homogeneous medium.** The emitters used to probe the local intensity in our ICSE experiment are commercially available DNA-based nanorulers decorated at their extremities with Atto 542 fluorescent molecules (see Methods). For the sake of simplicity, we will demonstrate our technique in the case where the spatial fluctuations of the fluorescence emission are solely determined by the local excitation intensity (see Methods). One key feature of our approach is that it does not require to directly image the nanoprobe, neither resolve the separated emitters: the properties of the internal light distribution are obtained by simply collecting the diffuse emission $I_{\text{f,tot}} = I_{\text{f,1}} + I_{\text{f,2}}$ from both ends of an isolated nanoruler. The non-imaging nature of the method makes it also free of common polarization-related artefacts[33]. Moreover, since the intensity correlation information obtained with our method is inherently related to a length scale set by $D$, the measurement is not affected by convolution artifacts nor by the diffraction limit. Figure 1a–d shows a few illustrative fluorescence patterns for the smallest nanoruler ($D = 30$ nm, Gatta-sted 30) embedded in a homogeneous agarose gel, with 3 of them clearly visible inside the field of view (see Methods). The intensity detected at the position highlighted by the red circle is shown in panel 1e for 500 independent realizations, resulting in the p.d.f. shown in panel 1f. Based on the illumination conditions (linearly polarized light, $\lambda_0 = 532$ nm, 0.75 NA objective), the expected speckle size $S$ inside the agarose mixture ($n \simeq 1.33$) is about 260 nm, much larger than the length $D$ of the nanoruler.

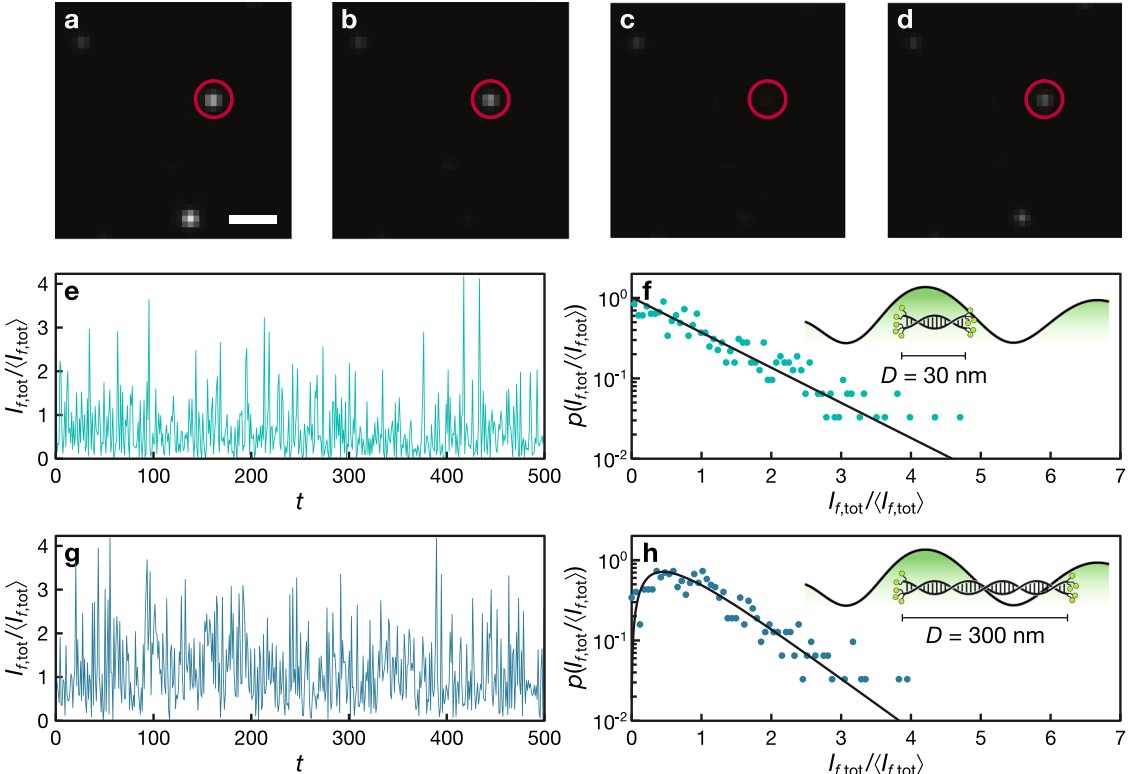

**Fig. 1 Fluorescence emission probability density from individual nanorulers. a–d** Fluorescence from three $D = 30$ nm Gatta-sted nanorulers under four different excitation patterns. The nanorulers are embedded into a transparent gel matrix (2% agarose in water). The white scale bar corresponds to 1 μm. **e, g** Total intensity $I_{tot}$ from nanorulers with $D = 30$ and 300 nm obtained for 500 different excitation patterns labeled with the index $t$. **f, h** Probability density function $p(I_{tot})$ for $D = 30$ and 300 nm. Fitting with Eq. (1) returns $N = 1.00 \pm 0.05$ and $N = 1.95 \pm 0.05$ for $D = 30$ nm and $D = 300$ nm, respectively.

We therefore expect that, on average, both extremities of the nanoruler will fall inside the same speckle grain and therefore probe the same field $I(\mathbf{r}_1) \simeq I(\mathbf{r}_2)$. This is confirmed by fitting the probability density function with Eq. (1), which returns $N = 1.00 \pm 0.05$.

A different fluctuation statistics is observed when repeating the experiment using a longer nanoruler ($D = 300$ nm, Gatta-sted 300). Now the fluorophores are placed at a distance $D > S$ and will therefore probe two independent intensity values, with a probability density function $p(I_{f,tot})$ resulting from the convolution of two p.d.f. distributions. Panels 1g, h show the measured intensity fluctuations and the associated p.d.f., returning $N = 1.95 \pm 0.05$.

When $D \approx S$, the value of $I_{f,1}$ is partially correlated to $I_{f,2}$ and the shape of $p(I_{f,tot})$ can still be modeled with Eq. (1) by simply allowing non integer $N$ values. A value of $N \simeq 1$ corresponds to a strongly correlated emission while a value of $N \simeq 2$ results form the sum of two uncorrelated sources. Thus, in general, the shape of the intensity p.d.f. (and hence the value of the $N$ parameter) is determined by the ratio of the distance $D$ to the intensity correlation length $S$. In the homogeneous case, this behavior can be compared with that obtained from a simple plane wave summation calculation (Fig. 2a and Methods).

Numerically obtained curves are compared to experimental data in Fig. 2b, showing good agreement when the refractive index in the calculation is taken to be $n = 1.33$ (corresponding to the actual refractive index of the water-agarose gel embedding the nanorulers). The inset reports the monotonic dependence of $S$ or, equivalently, $n$ as a function of $N$ for a Gatta-sted with $D = 90$ nm, showing that our approach can be used to retrieve either value in a dielectric medium even using a single $D$ value.

**Nanorulers in a turbid medium**. Having validated our approach in a homogeneous medium, we now apply it to a highly scattering material whose internal electromagnetic field distribution is inaccessible by traditional techniques. For this purpose, we prepared dense suspensions of ZnO nanoparticles in a water-agarose gel (see Methods) containing a dilute concentration of nanorulers. Keeping a low concentration of nanorulers allows to collect diffuse light originating from one Gatta-sted at a time (see Supplementary Fig. 2). On average, the nanorulers will be buried inside the scattering medium so that laser light will impinge on the fluorophores with all polarizations, forming an intensity landscape whose statistical properties are independent of the illumination optics[34]. Due to the presence of three independent polarization channels on each emitter, we expect that now $N$ will vary from 3 when $D \ll S$ to 6 for $D > S$, i.e., when probing two uncorrelated positions.

Experimental data for nanorulers with $D = 30$ and 160 nm fitted with Eq. (1) are shown in Fig. 3a, b. Since in this case the fluorophores lie in the close vicinity of scattering nanoparticles, significant near-field contributions are expected to affect the spatial intensity distribution which cannot be modeled by the simplified plane wave approach used for the homogeneous medium. Even when using the appropriate value for the effective refractive index of the inhomogeneous medium (see Methods), the such calculated dependence of $N(D)$ cannot reproduce correctly the experimental data (see Fig. 3c).

To gain more insight in the situation with scattering, we can make use of a rigorous approach based on generalized multi-particle Mie theory (GMMT, see Methods and Supplementary Fig. 3). Exploiting a recent implementation on graphical processing hardware, this method allows to solve Maxwell's

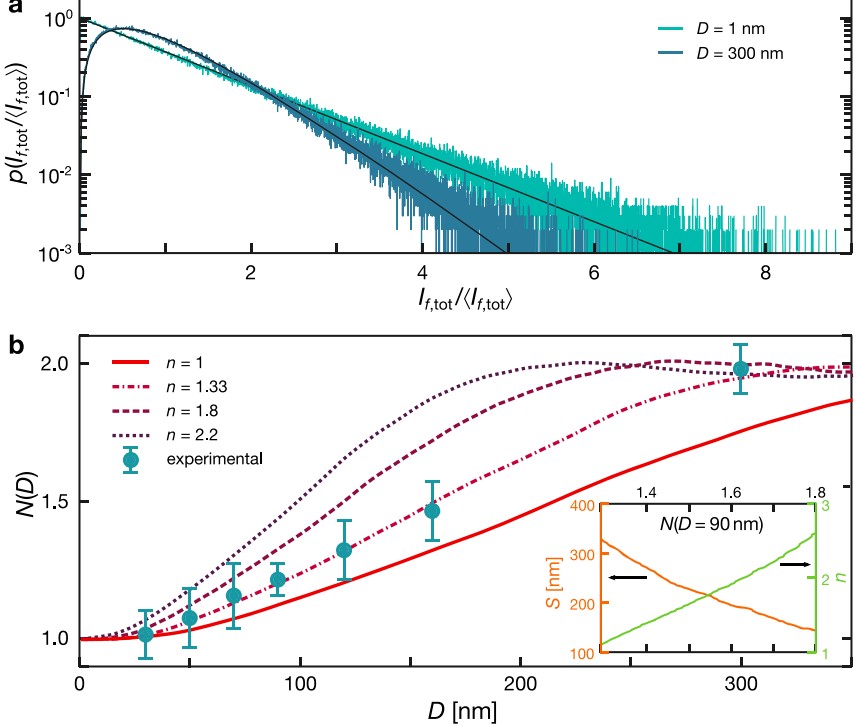

**Fig. 2 Dependence of the probability density of the intensity on nanoruler size: homogeneous medium. a** Numerically calculated probability density function obtained for $D = 1$ and 300 nm in a homogeneous medium with $n = 1.33$. Solid lines represent Eq. (1) with $N = 1$ and $N = 2$. **b** Numerically retrieved $N(D)$ curves for different refractive indices. The experimental data is shown as full circles with an uncertainty estimated by averaging the results over 12 different nanorulers embedded in two different samples, independently for each value of $D$. The inset shows the relation between $n$ and $S$ as determined numerically for a nanoruler of size $D = 90$ nm.

equations for mesoscopic media comprising hundreds of thousands of particles[35,36]. By calculating the full 3D intensity distribution inside a representative aggregate modeled after the experimental sample, it is possible to extract the values of $N(D)$ for synthetic nanorulers placed in small voids between the dense nanoparticle packing.

Results obtained using the rigorous method are in excellent agreement with the experimental data, without the need of any scaling parameter. The actual value of $N(D)$ is consistently lower than the simple plane wave prediction, ranging from $N \simeq 2.5$ to 5 rather than from 3 to 6 (Fig. 3c, dashed line). These results suggest that the presence of the scattering elements determines an effective reduction of the available polarization degrees of freedom. Indeed, at subwavelength distance from particle-void interfaces, the electromagnetic field is characterized by both propagating and evanescent components which could be either strongly enhanced or attenuated[37,38], thus decreasing the effective number of available degrees of freedom.

To further validate this hypothesis, we built a minimal configuration to probe the average orientation of the polarization ellipse at a position confined between two scattering particles (see Fig. 3d, e, Methods). Despite perfectly isotropic illumination conditions, the electric field polarization in the close proximity of these particles is clearly biased to rotate on a plane containing the inter-particle axis, signaling a local reduction of its orientation degrees of freedom.

In order to test the importance of polarization and near-field contributions—which are both taken into account in the numerical data but not in available theoretical models—we evaluate the normalized radial autocorrelation function $C_I(|\mathbf{r} - \mathbf{r}'|) = \langle \delta I(\mathbf{r}) \delta I(\mathbf{r}') \rangle$ from a set of 3D calculations relative to different isotropic illumination conditions The current reference model for spatial intensity correlations inside a 3D medium, initially

developed for scalar waves and later extended to the vector-wave case[14,39] can account only for far-field or "universal" intensity correlations, irrespective of the specific realization of disorder. The theory predicts short-range correlations decaying as $\text{sinc}^2(k\Delta r) \exp(-\Delta r/\ell_s)$. The exponential term can be neglected as the scattering mean free path $\ell_s$ in our system is of the order of $\sim 10\,\mu\text{m}$ (see Supplementary Fig. 4), and therefore much larger than the wavelength inside the medium. The predicted $C_I$ is in excellent agreement with the effective-medium plane wave model, but is much broader than that obtained via the GMMT calculations, which exhibit a half-width half-maximum of just 41.4 nm instead of 75.5 nm (Fig. 4). The same discrepancy is observed when comparing intensity correlations outside the scattering particles (which is what we have access to, experimentally) to the case of the homogeneous host medium (dashed curves). As a matter of fact, even restricting the analysis to the low-index host medium only, the observed width of 58.4 nm is still significantly narrower than that predicted for the whole effective medium.

## Discussion
From our analysis, a few aspects appear to determine the sub-diffraction narrowing of the short-range intensity correlations. The first is connected to the particle-like configuration introducing sharp intensity enhancements and jumps at the interfaces (the particles themselves are too small to sustain internal modes). At short distances, these features in the intensity pattern tend to squeeze the correlation peak toward the length scale of the particles, and their effect can be isolated by comparison to the autocorrelation curve calculated in the host medium only. This is illustrated by the short-range difference between the green curves of Fig. 4 even though, notably, a substantial agreement is recovered at larger distances. At length scales even smaller than the

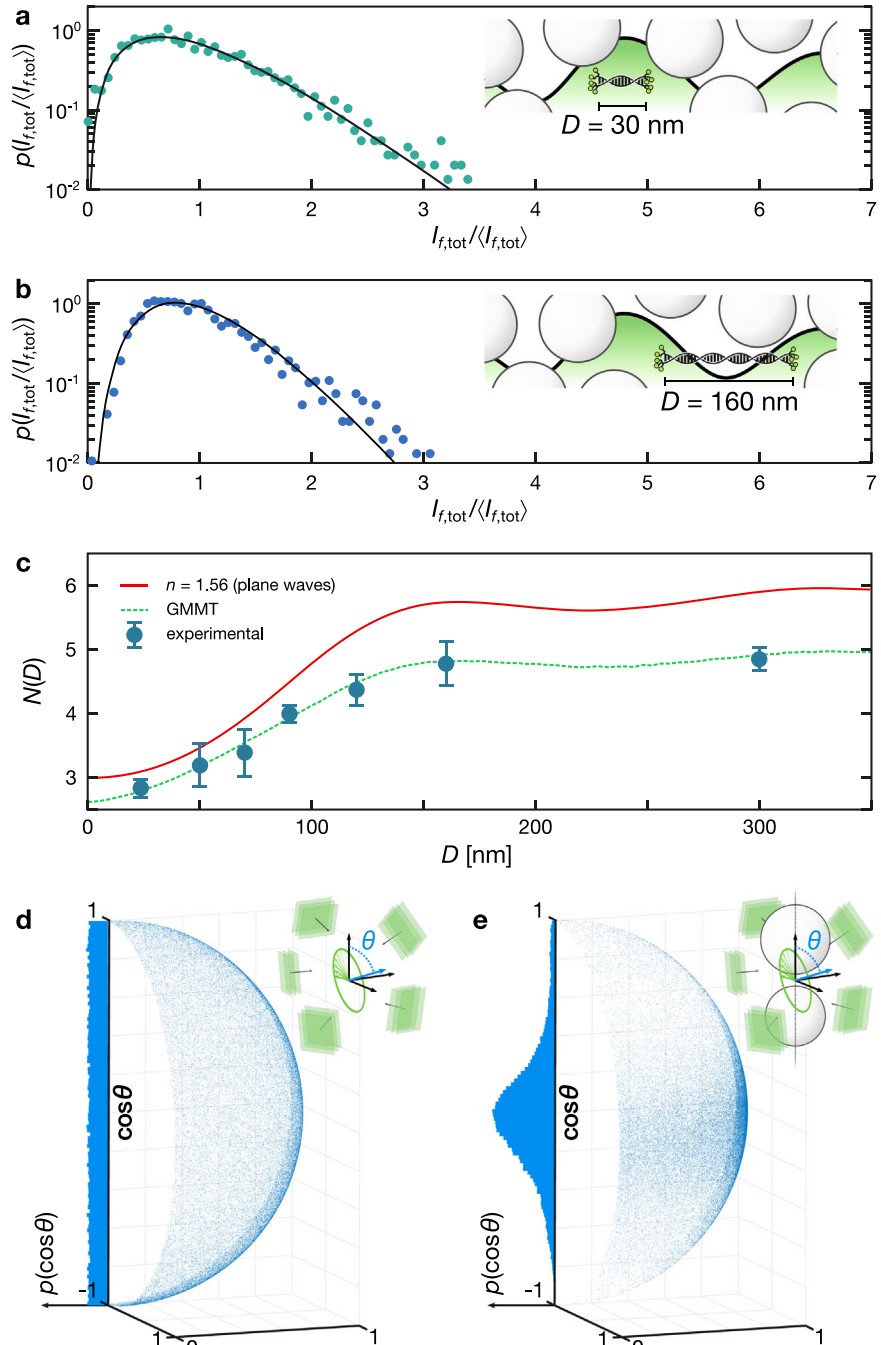

**Fig. 3 Dependence of the probability density function on nanoruler size: turbid medium. a**, **b** Experimental probability density function for $D = 30$ nm and 160 nm fitted with Eq. (1) (inset drawing not to scale). **c** $N(D)$ dependence obtained from a simple plane wave calculation (solid, red) and from generalized multiparticle Mie theory (GMMT) calculations (dashed, green). Full circles correspond to experimental data points, with an uncertainty estimated by averaging the results of 8 different nanorulers. **d**, **e** Average angular distribution of the unit vector normal to the polarization ellipse under isotropic illumination in a homogeneous effective medium and at a position between two scattering particles. The blue arrow in the sketches is parallel to the propagation direction and normal the polarization ellipse swept by the electric field (green arrow).

typical permittivity fluctuations $\langle \delta\epsilon(\mathbf{r})\delta\epsilon(\mathbf{r}')\rangle$, the nonuniversal character of $C_I$ is eventually expected to dominate. This is qualitatively confirmed by a comparison to a theoretical model for the "extreme near-field" regime[14] (see Methods).

A second aspect regards the role of correlations between orthogonal polarization components, a factor that is usually neglected in theoretical models[14]. Further analysis (see Methods) reveals a separate contribution that narrows the intensity

correlations and which vanishes when each polarization channel is considered separately. Remarkably, also this difference is only present when considering the intensity pattern across the whole (heterogeneous) medium, while it is absent both in the homogeneous case and when we limit the analysis to the host phase only.

Finally, it is worth commenting further on the oscillations of $C_I$, in particular their pitch, visible in the inset of Fig. 4 (see also

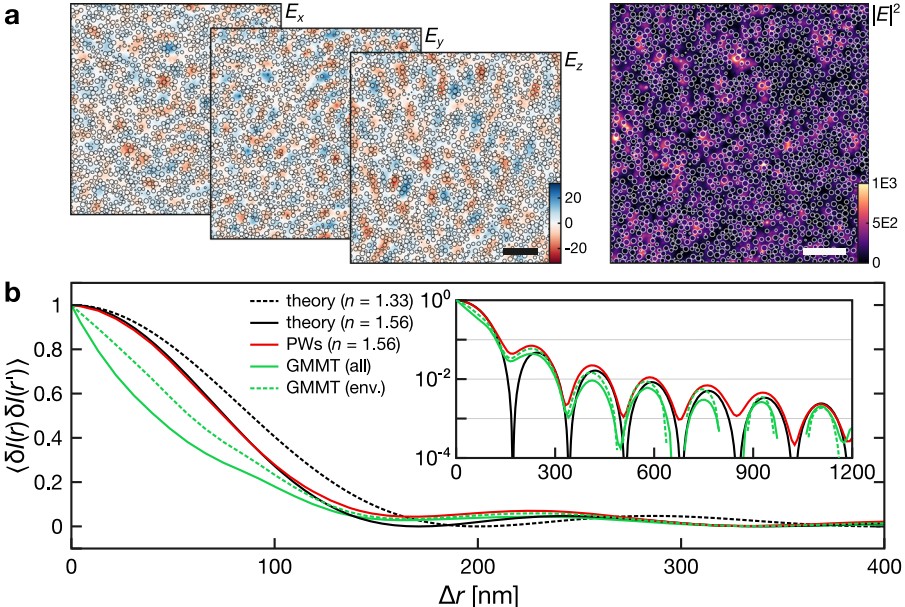

**Fig. 4 Autocorrelation of calculated near-field intensity maps. a** Color maps of both the electric field and intensity distribution in the $y = 0$ nm plane for a random illumination pattern. Scale bars represent $\lambda_0/n_{eff} = 341$ nm. **b** Radial average of the normalized autocorrelation profiles (semi-log scale in the inset). Colored curves correspond to: the homogeneous effective medium obtained by the summation of randomly oriented plane waves (PWs, solid, red), generalized multiparticle Mie theory (GMMT) calculations (solid, green), and GMMT calculations considering only host-medium regions (dashed, green). The black curves represent theoretical predictions for homogeneous media with effective (solid) and host (dashed) refractive index.

Methods). The period and position of the correlation peaks remains unaffected whether we consider the whole medium or only the host medium, hinting at the possibility to probe experimentally the effective permittivity of an heterogeneous medium even by having access to just one of its constituent phases. Being able to determine how electromagnetic intensity distributes into the different phases of a heterogeneous medium is a relevant concept both for sensing applications (an analyte can typically access only the voids of a nanophotonic structure) and to test the validity of effective-medium theories e.g., in the presence of resonances or near-field effects. Notably, these $C_I$ oscillations are mirrored exactly (with a phase shift of $\pi$) in the $N(D)$ curves, meaning that the information they encode is in principle experimentally accessible by measuring the ICSE, as we described in this paper. It is worth noting that, due to the significant near-field contribution determining all previously described effects, samples comprising particles with different shape, size, chemistry or arrangement may exhibit different types of deviations. In this respect, performing an ICSE characterization of nanostructured materials may allow to vary their microscopic details in order to induce tailored constraints on the extent of internal intensity correlations and their average polarization content.

In summary, we present an experimental study, supported with numerical calculations, on the correlations of light intensity and polarization inside complex photonic structures. The experimental method—also introduced in this work—makes use of DNA-based nanorulers, is not limited by diffraction or convolution artifacts, and can be applied to any type of bulk material. We analyse spatial correlations of the intensity on length scales that go well below the diffraction limit—thereby including near-field and polarization effects—and reveal several unexpected properties of the complex behavior of light inside disordered materials. There is a new range of previously inaccessible physical phenomena to be explored, most of which are lost in the propagating fields on the outside of the material and hence cannot be investigated by conventional experimental techniques. We believe

that the possibility to probe the behavior of light inside bulk nanostructured systems represents a significant leap forward, providing a new tool to help the development of photonic materials and devices, and at the same time explore the so far uncharted territory of nanoscale light-matter interactions inside complex 3D media.

## Methods

**Experimental apparatus.** Light generated by a low noise stabilized, single mode laser (Azure Light, 0.5 W, 532 nm) impinges on a computer-controlled Digital Micromirror Device (DMD, Vialux V-7000, 1024 × 768 pixels, 13.7 μm pitch), and is directed by a two lens system to the sample. An iris selects a specific area of the Fourier space of the DMD to avoid inhomogeneous contributions form high-intensity diffraction orders[40]. The DMD is modulated over an active area of 100 × 100 pixels with a random binary mask and imaged onto the sample. Light from a single Gatta-sted (whose concentration is tuned to have at most one nanoruler in the field of view) is collected with an Olympus UplanSApo ×100 objective, filtered through a dichroic mirror and imaged on a single-photon camera (Evolve Delta 512, pixel size 16 μm). A 3-axis piezometric actuator (Thorlabs MAX311D/M) ensures steady positioning of the sample.

In our measurement protocol, the sample is illuminated with a random DMD mask while acquiring a frame on the camera (200 ms exposure time). To avoid spurious effects due to stochastic fluctuations in the molecules photophysics, the response to same train of random masks is averaged over ten identical repetitions. The process is repeated 2500 times for different and uncorrelated DMD masks to collect a statistical ensemble of intensity values corresponding to 2500 different illumination conditions. The first moment μ of the intensity fluctuations is then calculated from the ensemble to obtain the normalized intensity.

**Sample properties and preparation.** The DNA-Origami technology from GAT-TAquant GmbH provides a choice of rigid and robust DNA-based nanostrands[41] decorated at their extremities with clusters of Atto 542 fluorescent molecules. For the STED variant used in our experiment, GATTAquant nanorulers have ~15–20 randomly oriented dye-molecules per fluorescent mark, which allows to get an average fluorescence read-out that is proportional to the local intensity irrespective of its polarization, and with a negligible local perturbation the intensity. Gatta-sted nanorulers are available off-the-shelf with lengths $30 \le D \le 300$ nm. Larger $D$ can be obtained with the same DNA-origami technique. Conversely, given their minimum size, a full scan of the spatial dependence of intensity correlations (i.e., including the fully correlated case) can be performed only if their typical length scale is above 30 nm. Another limitation of this cheap and ready-to-use approach is that the investigated sample must be kept hydrated (hence the aqueous agarose

gel), which limits the refractive index contrast of the scattering medium. This limitation however could be potentially surpassed by realizing a solid state nanoruler[42]. Moreover, compared to a fluorescent molecule, a solid state emitter could also withstand higher excitation intensities, allowing faster measurement times and, therefore, the possibility to study the dynamics of speckle field correlations in slowly diffusing media.

A 10 mM salt solution ($MgCl_2$ in $H_2O$) is prepared and stirred with 2% vol of low gelling temperature Agarose powder, (Sigma-Aldrich A9539). The Gatta-sted/Agarose solution is obtained by mixing 5 μL of the Gatta-sted solution with 50 μL of Agarose solution. The homogeneous sample is obtained by drop-casting 1 μL of the Gatta-sted/Agarose solution on a microscope coverslip, which is then sealed under a second coverslip using transparent nail polish.

For the scattering sample, a 4 M solution of ZnO nanoparticles (Sigma-Aldrich 544906, particle radius 42 nm with 10% polydispersity[43]) in water is prepared. 1 μL of the nanoparticle solution is drop-cast together with a 1 μL Gatta-sted/Agarose solution directly on a microscopy coverslip mixing it manually with the pipette tip. After evaporation of the excess water, the scattering gel is covered with a microscope coverslip and sealed with transparent nail polish. The resulting samples have a typical thickness comprised between 60 and 100 μm, with buried nanorulers being typically measured from an intermediate depth within the scattering slab. Multiple samples are prepared for each value of $D$. The final ZnO volume fraction obtained for scattering samples is found to be around $(38 \pm 2)$% for all samples, as estimated via a repeated weighting method in cuvettes containing 1–4 mL of solutions (ZnO density of 5.6 g cm$^{-3}$). The resulting scattering strength (see Supplementary Fig. 4) corresponds to a value of $k\ell \geq 150$, a regime where the local density of states is known to be spatially invariant for this type of scattering system[19]. Similarly, the non-absorbing, all-dielectric nature of the investigated samples avoids the possibility of spatially varying fluorescence quenching, which simplifies the analysis of the experimental results, while the extremely high quantum yield (93%) of the Atto 542 emitters makes their emission rates largely unaffected by different environmental conditions such as the presence of hydrophobic materials[44] (see also Supplementary Fig. 5). Under these circumstances, the observed fluorescence fluctuations are determined solely by the excitation rate and thus represent a direct measurement of the local optical intensity. In the more general case of samples with spatially varying density of states or non-radiative quenching terms, additional information regarding the magnitude of each component could be obtained by, e.g., using multiple excitation wavelengths, nanorulers decorated with different emitters, or by measuring the local density of states at each emitter position.

**Numerical calculations: homogeneous medium.** Numerical calculations in the homogeneous and in the effective-medium case are performed by summing $10^3$ plane waves with random amplitude (drawn uniformly between 0 and 1), random propagation direction, random polarization direction and random phase. The resulting p.d.f. are then averaged over $10^4$ realizations of the random wave parameters. For each realization, we plot the obtained 3D speckle realization over a linear segment representing a Gatta-sted and record the sum of the intensity values at its extremities for different $D$ values. The probability density function obtained from this ensemble is fitted with a one-parameter function $p(x) = N^N/\Gamma(N) x^{N-1} \exp(-Nx)$ to retrieve the average number of polarimetric degrees of freedom for each $D$ value.

Plane wave calculations require an effective refractive index as an input parameter to rescale the wavelength inside the medium. For the water/Agarose gel, we estimated the $n_{gel} \simeq n_{H_2O} = 1.33$. As regards the ZnO/gel sample, feeding values of $n_{ZnO} = 1.96$ and ZnO volume density of 38% into either Bruggeman's or Maxwell-Garnett mixing formulas returns in both cases $n_{eff} = 1.56$. Fitting curves shown in Fig. 2 are obtained assuming a homogeneous medium with $n_{gel} = 1.33$ returned $N = 1.00 \pm 0.03$ and $N = 1.94 \pm 0.01$ for $D = 1$ and 300 nm, respectively.

The 3D orientation of the polarization ellipse for the homogeneous case has been obtained either by summing $5 \times 10^3$ plane waves as specified above, or by generating directly a synthetic value of the electric field by drawing $3 + 3$ Gaussian random variates for the real and imaginary part of each field component. This method, which represents a 3D generalization of a previous study on 2D images[45], corresponds to summing infinitely many plane waves and returns a distribution that is indistinguishable from that obtained after summing ~100 plane waves manually.

**Numerical calculations: turbid medium.** A disordered, spherical packing of non-overlapping spherical particles with a polydisperse distribution of radii (uniform between 38 and 46 nm) is generated using a molecular dynamics approach[46]. An initial loose packing is compressed until the experimentally determined volume density of 38% is reached. The final packing consists of a spherical aggregate with a diameter of 5 μm comprising ~$1.2 \times 10^5$ particles. In a subsequent step, $10^4$ rigid segments of length $D$ are randomly placed inside the aggregate ensuring that they do not overlap any particle, to mimic an ICSE measurement (see Supplementary Fig. 3). Full-wave calculations are performed based on GMMT which conveniently allows to model light scattering from large aggregates of spherical particles with high efficiency thanks to hybrid computing platforms. Considering the ZnO/gel permittivity contrast

and the average size of the nanoparticles (size parameter $x = kr \simeq 0.0157$) multipolar expansions for GMMT calculations are truncated at $l_{max} = 3$.

To mimic the quasi-isotropic illumination conditions associated to a diffusive medium, near-field distributions at the segment positions are calculated by summing different independent calculations performed for 250 incoming plane waves with random amplitude (drawn uniformly between 0 and 1), random propagation direction, random polarization direction and random phase. In addition to the segments, we compute the near-field distribution also at the $y = 0$ nm plane with a spatial resolution of 4 nm to estimate the intensity correlation length from the half-width half-maximum of the autocorrelation (AC) peak (see Fig. 4). Separate AC curves are eventually obtained considering either the full intensity map or only the values outside the ZnO particles.

The 3D orientation of the polarization ellipses has been calculated at the middle point of the gap between two representative scattering particles of radii 40 and 44 nm placed at a typical interdistance $\rho^{-1/3} \simeq 92.5$ nm, with $\rho$ as the particle number density. Here, $10^5$ different illumination conditions have been stored and combined to form isotropic illumination conditions with different dephasing vales and random amplitudes. Despite this, the resulting light fields at the mid-gap position retains a clear preference for the normal direction to the polarization ellipse to lie on the plane perpendicular to the cylindrical axis of the two-particle system, signaling the presence of a constraint on the degrees of freedom of the associated electric field.

**Universal and nonuniversal intensity correlations: theoretical models.** Theoretical models describing intensity–intensity correlations in disordered media are typically based on mesoscopic regime assumptions that are valid when $\ell_c \ll \lambda_0/n_{eff} \ll \ell_s$. Here, $\ell_c \sim 23$ nm represents the typical length scale of permittivity fluctuations in our sample, as estimated from a Gaussian fit of its autocorrelation curve (see Supplementary Fig. 6a), while $\ell_s$ is the scattering mean free path which for our material is of the order of a few μm (for reference, the independent scattering approximation, which can be taken as a lower bound for $\ell_s$, predicts ~4 μm at our particle density). Comparing these values to $\lambda_0/n_{eff} = 341$ nm shows that our experimental configuration satisfies the mesoscopic assumptions by at least an order of magnitude in each direction.

The main model accounting for the universal part of intensity correlations can be derived following a diagrammatic approach in the far-field approximation[39,47] and assuming unpolarized light[48], i.e., $\langle E_m(\mathbf{r})E_n^*(\mathbf{r}')\rangle = \delta_{mn}$. Following this approach, one eventually obtains the expression $C_I(\Delta r) = \text{sinc}^2(k\Delta r) \exp(-\Delta r/\ell_s)$ for the normalized intensity correlations, which is a universal result for both scalar and electromagnetic waves. As shown in Supplementary Fig. 6a, this model provides an excellent description of intensity correlations starting from $\Delta r \geq \lambda_0/n_{eff}$ while it fails at shorter distances where nonuniversal contributions are naturally expected to play a significant role.

In our experimental configuration, a typical length scale for inter-particle distances can be estimated in the order of $\rho^{-1/3} \simeq 92.5$ nm, leaving an average gap width between particles <10 nm and therefore an even smaller average distance from any dye molecule to the nearest particle interface. In this regime, it is safe to assume that the points where the intensity is probed are on average at a distance from the scattering elements that is much smaller than the correlation length $\ell_c$ itself. To the best of our knowledge, a complete description of intensity–intensity correlations has not been developed yet for this extreme near-field regime inside a scattering medium.

Nonetheless, a simple theoretical model is available for near-field correlations in close proximity to the external interface of a scattering medium, arguably because this has been so far the only quantity that was experimentally accessible[48]. In this configuration, intensity correlations are expected to exhibit a nonuniversal decay as $M^2(3/2, 1, -\Delta r/\ell_c)$, where $M$ is the hypergeometric function[49]. As can be seen, the hypergeometric model predicts a much narrower intensity correlations length, in agreement with the numerically determined $C_I$ curve up to a distance of a few nm.

As we discussed, another aspect connected to the non-universality of the local environment where the intensity is probed, is that the permittivity configuration becomes inhomogeneous and anisotropic at such small scales. To highlight the presence of polarization-related effects, we compare in Supplementary Fig. 6b the intensity correlation curves calculated either as $\langle \delta I(\mathbf{r})\delta I(\mathbf{r}')\rangle$ (i.e., as we have done throughout the paper), or as $\sum_{i=1}^{3}\langle \delta I_i(\mathbf{r})\delta I_i(\mathbf{r}')\rangle/3$ (i.e., manually obliterating any possible cross-component intensity correlation). As can be seen, the obtained correlation curves are indistinguishable except for the intensity distribution over the whole scattering medium, i.e., both inside and outside the particles. In this case, accounting correctly for polarization leads to a narrowing of the correlations.

Finally, Supplementary Fig. 6c shows in more detail the comparison between numerical $C_I$ curves and the far-field model obtained for effective refractive index values of 1.56 and 1.33. As can be seen, the far-field behavior of the 1.33 case is incompatible with all curves, including that obtained when considering intensity fluctuations only in the aqueous environment (which occupies >60% of the total volume) where the scattering particles are immersed. This suggests that, even when probing intensity correlations only outside high-permittivity particles, long-range

$C_I$ oscillations are still aware and affected by the presence of the particles and their refractive index value.

## Data availability

The data that support the findings of this study are available from the authors upon reasonable request. MATLAB scripts used to generate and process numerical simulation data are publicly available at https://github.com/lpattelli/NCOMMS-20-45047A.

## Code availability

The GMMT code is freely available at https://github.com/disordered-photonics/celes (release version 2.2). With respect to third party particle packing codes, we refer to the original publication[46]. Routines and scripts for the analysis of experimental and numerical data are available from the authors upon reasonable request.

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

## Acknowledgements
M.L. acknowledges "Fondazione CON IL SUD", Grant "Brains2south", Project "Localitis". L.P. acknowledges Progetto Premiale MIUR "Volume photography" and NVIDIA Corporation for the donation of the Titan X Pascal GPU used for this research. L.P. wishes to thank Amos Egel, Alice Boschetti, and Francesco Riboli for fruitful discussion.

## Author contributions
M.L. and L.P. equally contributed to the results presented in this paper. M.L. conceived and realized the experiment and the measurements. L.P. conceived and performed the numerical simulations and data analysis. S.D.P. helped with the optical characterization of the samples. M.L. and L.P. wrote the paper, with feedback from S.D.P., D.S.W., and G.R. All authors contributed to the interpretation of the results.

## Competing interests
The authors declare no competing interests.
