## [Peer Review File · Nature Communications]

REVIEWER COMMENTS

Reviewer #1 (Remarks to the Author):

The paper by Leonetti et al. aims to characterize spatial intensity fluctuations, known as speckle, inside a three-dimensional random medium. To this end, the authors propose and design an experiment using commercial DNA-origami nano-spacers with fluorophores attached at each end. By tuning the distance between the fluorophores from being smaller to being comparable to a speckle size S , they hope to measure the intensity distribution function. The experiment is similar to measuring the $P(I)$ for a far-field speckle with a detector covering an increasing size area. J.W. Goodman, see their ref. [28] has described the underlying theory (originally in the 1970s). For a finite-size detection area, $P(I)$ deviates from a negative exponential and can be modeled with the so-called gamma distribution [their Eq.(1)]. Skipetrov et al.

(doi.org/10.1364/OE.18.014519, Optics Express 18, 14519-14534 (2010)) also discuss the relationship with the spatial autocorrelation function $g_2(r-r')$, a study the authors may find useful.

In my view, what the authors suggest is a novel and exciting new approach leveraging advances in DNA nanotechnology and fluorescent microscopy. The authors have designed the experiment very well, and the calibration data taken on a non-scattering sample, and presented in Fig 1 and 2, is precise and convincing. Moreover, the paper is clear and well written. Up to this point, I have no criticism. My main concern is the interpretation of the data taken in the scattering medium. To this end, the authors must discuss whether the presence of the ZnO nanoparticles can affect the brightness of the fluorophores. It is known that the fluorescence rate of a single molecule varies as a function of its distance from a nanoparticle; see e.g. the highly cited work of P Anger, P Bharadwaj, L Novotny PRL 96 (11), 113002, 2006. If the brightness of the fluorophore is affected this way, this would profoundly influence their measurement. See e.g. also Scalia et al., Optics Express 16; 23, 29342-52 (2015). This proximity effect could lead to additional spatial fluctuations of emitted intensity, even in the presence of a homogenous excitation light field on the length scale of the probe. It could well explain the observed smaller than expected size of the speckle. If that were the case, the authors would unfortunately not measure the actual bulk speckle with their technique. In my view, this wouldn't be very surprising since the fluorescent emission is known to be affected easily by the dielectric and chemical environment. Changes in fluorescent emission frequently appear even in the absence of Anderson localization or bandgap formation, which is widely exploited in FLIM (fluorescent lifetime imaging). A way to test whether this is the case would be to study different types of nanoparticles. After all, the bulk scattering speckle should not be affected by the type of nanoparticle used.

In my view, the authors also must address the following questions:

- 1) Data from the calibration sample is shown but not from the actual turbid sample of interest; the authors must add images similar to Fig 1a)-d) for the reader to get an idea about the measurement's accuracy and sensitivity.
- 2) The authors must clearly state the optical density of the sample, L/l_s . l_s is mentioned to be around 10 microns, so I assume L/l_s is larger than ten?

Reviewer #2 (Remarks to the Author):

In the present manuscript, measurements are presented that reveal the intensity correlations of light inside a scattering medium. The authors use two fluorescent markers spaced by a rigid molecule to probe the statistics of the sum of the intensity at two points separated by tens to hundreds of nm. In a synthetic speckle field inside a clear medium, they find the expected behavior, namely that at a short distance the total intensity follows purely exponential statistics while at a larger distance the statistics become that of the sum of two independent speckle intensities.

In a medium loaded with scattering particles, a drastic change of the statistics is found, for a large part due to polarization scrambling by the scattering particles. Numerical calculations based on multi-particle Mie theory reproduce the experimental results well.

The work presents a new method and new insights in field correlations inside scattering media, considerably improving the state of knowledge from the existing oversimplified theory.

There are a few comments that I would like the authors to respond to before publication.

- On p4, ... our approach can be used to retrieve both n and S ... The "both" is confusing as n and S are dependent. If this is indeed the case, it should be phrased as " n , or equivalently, S " or similar. One cannot retrieve two independent quantities from a single scalar-valued measurement.
- While the measurement method employed by the authors can be applied to a reasonably general range of media, the results are extremely dependent on the exact composition of the random medium, i.e., the results are very unlikely to carry over to different random media. Moreover, the method only measures the correlations restricted to one component (the watery gel).
- Is it in principle possible to perform the same method with freely diffusing molecules, e.g. in tissue? This would be a very interesting prospect indeed.
- What is the shape of the ZnO nanoparticles? From the manufacturer's information, I do not get the impression that they should be spherical. Even for non-spherical particles, one can expect reasonable agreement with Mie theory in the far field [M. I. Mishchenko, "Light scattering by size-shape distributions of randomly oriented axially symmetric particles of a size comparable to a wavelength," Appl. Opt. 32, 4652–4666(1993).], does this reasoning carry over to near field effects?
- The depictions in the insets of Fig 3a,b seem a bit misleading as the particles should be much larger?
- Could extra information be extracted using time-correlated single-photon counting and/or multiple excitation wavelengths?

In light of the importance and originality of the manuscript, I advise publication of the manuscript after the authors have carefully considered the above remarks.

We thank both Reviewers for their careful Referral. We gladly acknowledge that both find the work very interesting. We believe that the suggestions and feedback that they provided allowed us to improve the manuscript and its clarity.

Reviewer #1 (Remarks to the Author):

The paper by Leonetti et al. aims to characterize spatial intensity fluctuations, known as speckle, inside a three-dimensional random medium. To this end, the authors propose and design an experiment using commercial DNA-origami nano-spacers with fluorophores attached at each end. By tuning the distance between the fluorophores from being smaller to being comparable to a speckle size S , they hope to measure the intensity distribution function. The experiment is similar to measuring the $P(I)$ for a far-field speckle with a detector covering an increasing size area. J.W. Goodman, see their ref. [28] has described the underlying theory (originally in the 1970s). For a finite-size detection area, $P(I)$ deviates from a negative exponential and can be modeled with the so-called gamma distribution [their Eq.(1)]. Skipetrov et al. (doi.org/10.1364/OE.18.014519, Optics Express 18, 14519-14534 (2010)) also discuss the relationship with the spatial autocorrelation function $g_2(r-r')$, a study the authors may find useful.

We thank the Reviewer for their careful analysis of our paper, and the interesting reference suggestion (now included as ref. [9] in the main text).

In my view, what the authors suggest is a novel and exciting new approach leveraging advances in DNA nanotechnology and fluorescent microscopy. The authors have designed the experiment very well, and the calibration data taken on a non-scattering sample, and presented in Fig 1 and 2, is precise and convincing. Moreover, the paper is clear and well written. Up to this point, I have no criticism.

We thank the Reviewer again for the very positive evaluation of our approach and experimental design.

My main concern is the interpretation of the data taken in the scattering medium. To this end, the authors must discuss whether the presence of the ZnO nanoparticles can affect the brightness of the fluorophores. It is known that the fluorescence rate of a single molecule varies as a function of its distance from a nanoparticle; see e.g. the highly cited work of P Anger, P Bharadwaj, L Novotny PRL 96 (11), 113002, 2006. If the brightness of the fluorophore is affected this way, this would profoundly influence their measurement. See e.g. also Scalia et al., Optics Express 16; 23, 29342-52 (2015).

This proximity effect could lead to additional spatial fluctuations of emitted intensity, even in the presence of a homogenous excitation light field on the length scale of the probe. It could well explain the observed smaller than expected size of the speckle. If that were the case, the authors would unfortunately not measure the actual bulk speckle with their technique. In my view, this wouldn't be very surprising since the fluorescent emission is known to be affected easily by the dielectric and chemical environment. Changes in fluorescent emission frequently appear even in the absence of Anderson localization or bandgap formation, which is widely exploited in FLIM (fluorescent lifetime imaging). A way to test whether this is the

case would be to study different types of nanoparticles. After all, the bulk scattering speckle should not be affected by the type of nanoparticle used

This is definitely one of the key aspects of our work and it clearly deserves to be discussed in more detail in this reply and the manuscript. As pointed out by the Reviewer and explained in the relevant paper by Novotny and co-workers (now cited as ref. [29]), fluorescence emission of a molecule in a nanostructured material is affected by the excitation efficiency, the spontaneous decay rate and the nonradiative decay rate. In principle, our investigation technique can be used to probe the combined effect of all three terms. In this work, we demonstrated our technique in the simple yet common case where the latter two terms can be assumed to be independent from the position inside the sample (see next paragraph). In this circumstance, observed fluorescence fluctuations will be determined solely by the excitation efficiency and be a direct measurement of the local optical intensity. In the more general case, additional information regarding the magnitude of each component could be obtained by, e.g., using multiple excitation wavelengths, nanorulers decorated with different emitters, or by measuring the local density of states at each emitter position.

To understand why, in our present case, it can be safely assumed that the spontaneous decay rate of our samples is constant throughout their volume, we can refer to the results published by R. Sapienza and co-workers in [PRL 106, 163902 (2011)] (cited in the manuscript as ref. [18]) for a very similar sample also made of ZnO nanoparticles with embedded emitters. Therein, by varying the average particle size and the contrast with the host medium, the authors were able to obtain extremely high scattering strengths of $k\ell = 9.4$ and $k\ell = 24$, with ℓ being the transport mean free path. What they showed is that, when the scattering strength decreases to $k\ell = 24$, the distribution of observed decay rates (i.e., local density of states) becomes narrowly peaked around a constant value which is independent of the position inside the medium. In our work, we have studied a scattering sample with only moderately high scattering strength. Using our estimated *scattering* mean free path value of $8 \mu\text{m}$ (the transport mean free path will be even larger) we obtain $k\ell \geq k\ell_s \sim 150$, which is much weaker than the least scattering sample of Sapienza et al., thus ensuring that the observed decay rates are independent of the fluorophore position with respect to our scattering nanoparticles.

As regards the comparison between our work and the paper by Novotny and co-workers, the following additional considerations also concur to support our conclusions:

1) nanorulers suitable for our experimental technique are decorated with several randomly-oriented molecules (15-20) at each end. This ensures that the possible presence of fluorescent emission fluctuations is averaged out among different emitters experiencing slightly different environmental conditions. This is a significant difference with the effects described in [PRL 96, 113002 (2006)], which are observed in a single-molecule regime.

2) The quenching effect observed in [PRL 96, 113002 (2006)] is due to the ohmic losses induced by a metal particle. Conversely, we have studied a non-absorbing, all-dielectric scattering nanostructure. We agree with the Reviewer that applying our technique to a plasmonic 3D system would require a more complex interpretation of the data, as discussed above. Moreover, plasmonic systems can exhibit intensity hotspots with extremely small volumes, which may be too small to be probed with DNA nanorulers. Therefore at the moment, the application of our technique to metallic nanostructures remains beyond the

scope of our work. We have now specified this limitation explicitly in the manuscript, based on the minimum length of available nanorulers.

A second reference cited by the Reviewer is that of Scalia and co-workers, which studies effects on the fluorescence rate of low-quantum-yield emitters due to the different wettability properties of scattering inclusions or their volume.

The effects reported by G. Scalia et al., although very interesting, to our comprehension do not apply to the present case nor affect the general validity of our technique for the following reasons:

1) As explained by G. Scalia et al., the observed deviations occur only for emitters with a low yield. No significant effect has been observed for an emitter with a higher yield (Alexa 488, QY 92%). In our experiment, we have used as emitter Atto 542 (the previous version of the manuscript erroneously reported Alexa 532, and we apologize for this typo) which has a spectral overlap and QY that are even larger than those reported for Alexa 488 (93%), thus ensuring a high versatility and robustness for our designed application. This point is now explicitly mentioned in the methods of the revised manuscript to provide guidance to potential users of our technique when choosing appropriate emitters for their specific application.

2) Moreover, G. Scalia and co-workers conjecture that the fluctuations observed for low-QY emitters are due to an “hydrophobic effect on the structure of water” in the vicinity of nanoparticles. Even though we used high-QY emitters, we investigated the possible presence of this effect by measuring the gattaquant fluorescence emission intensity as a function of the distance from an hydrophobic surface. In detail, we analyzed a sample made of nanorulers ($D=160$ nm Gattaquant) dispersed in a water-agarose gel matrix over a hydrophobic PDMS layer. Measurements have been performed with an Olympus FV1200MPE confocal microscope in reflectance configuration and with z-resolution of 150 nm (smaller than marker size in the graph). Individual nanoruler emission intensity has been extracted by integrating counts in 180 nm square region of interest at the z value for which the nanoruler is in focus.

The new measurement confirms the absence of this alleged effect in our sample. A new figure in the SI shows experimental data confirming how high-yield emitters are unaffected by the proximity to hydrophobic surfaces:

3) Importantly, the systematic variations observed in [Optics Express 16, 29342 (2015)] for low-yield emitters appear to depend on the particle concentration and chemical composition while the authors do not report “single particle” measurements. It seems therefore plausible

to assume that these average properties affect all fluorophores equally in the suspension, independently of their position. Given that our technique relies only on the normalized statistical distribution of the observed emission rate, it would still be applicable to different samples separately (average sample properties would affect the emitters at both ends of the nanorulers equally), and thus provide results that are physically significant and that directly comparable even between different samples.

Finally, as regards the dependence of bulk speckle properties on the type of nanoparticles used, their positions, spatial correlations, etc. – we believe that this is a quintessential example of a non-universal property due to its inherent near-field nature. Indeed, as also reported in [31,47], it could well be the case that the spatial extent of speckle correlations is affected by the microscopic details of the sample and we believe that this could be an interesting open question to address in the future using our newly developed technique. We still thank the Reviewer for raising this point, as we have now made it clearer in the text that the peculiar effects that we reported for the scattering configuration are not necessarily expected to hold when changing either the mesoscopic or microscopic details of its structure.

Changes to the manuscript:

A new Figure S5 was added to the SI. The following paragraphs were added to the main text:

page 3:

“For the sake of simplicity, we will demonstrate our technique in the case where the spatial fluctuations of the fluorescence emission are solely determined by the local excitation intensity (see methods)”

page 8, “Sample properties and preparation” paragraph:

“Gatta-sted nanorulers are available off-the-shelf with lengths $30 \leq D \leq 300$ nm. Larger D can be obtained with the same DNA origami technique. Conversely, given their minimum size, a full scan of the spatial dependence of intensity correlations (i.e., including the fully correlated case) can be performed only if their typical length scale is above 30 nm”

“The resulting scattering strength (see Figure S4 in the SI) corresponds to a value of $k\ell \geq 150$, a regime where the local density of states is known to be spatially invariant for this type of scattering system [18]. Similarly, the non-absorbing, all-dielectric nature of the investigated samples avoids the possibility of spatially varying fluorescence quenching, which simplifies the analysis of the experimental results, while the extremely high quantum yield (93%) of the Atto 542 emitters makes their emission rates largely unaffected by different environmental conditions such as the presence of hydrophobic materials [43] (see also Fig. S5 in the SI). Under these circumstances, the observed fluorescence fluctuations are determined solely by the excitation rate and thus represent a direct measurement of the local optical intensity. In the more general case of samples with spatially varying density of states or non-radiative quenching terms, additional information regarding the magnitude of each component could be obtained by, e.g., using multiple excitation wavelengths, nanorulers decorated with different emitters, or by measuring the local density of states at each emitter position.”

In my view, the authors also must address the following questions:

1) Data from the calibration sample is shown but not from the actual turbid sample of interest; the authors must add images similar to Fig 1a)-d) for the reader to get an idea about the measurement's accuracy and sensitivity.

We have added a new Figure S2 in the Supporting Information showing the fluctuating emission intensity from a buried molecule for different speckle configuration.

2) The authors must clearly state the optical density of the sample, L/l_s . l_s is mentioned to be around 10 microns, so I assume L/l_s is larger than ten?

We have estimated experimentally the optical density of the sample by measuring the scattering mean free path of ZnO-agarose mixtures of different thicknesses with an approach similar to M. Kempe et al. *JOSA A* 14.1 (1997): 216-223. The result is consistent with a scattering mean free path of 10 μm as initially postulated. A new figure reporting the measurements and measurement details is present in supplementary materials.

The resulting optical density of our samples is therefore of the order of 10 as assumed by the Reviewer, with buried nano rulers being typically measured from an intermediate depth within the scattering slab. Nonetheless, we should note that our technique allows to probe

3D spatial correlations for diffuse-like illumination even when the optical density of the investigated sample is not in the diffusive regime. In fact, in an ICSE experiment, the nanorulers are already illuminated with a fully developed speckle pattern (see Figures 1 and 2) generated by the random configurations of the spatial light modulator, akin to that obtained by a scattering medium deeply in the diffusive regime, while the contribution of near-field fluctuations to the spatial extent of 3D speckle fluctuations (which we are able to measure for the first time) develop only in close proximity to each scattering element, irrespective of the optical depth at which the measurement is performed.

Changes to the manuscript:

page 8:

“The resulting samples have a typical thickness comprised between 60 and 100 μm , with buried nanorulers being typically measured from an intermediate depth within the scattering slab”

Reviewer #2 (Remarks to the Author):

In the present manuscript, measurements are presented that reveal the intensity correlations of light inside a scattering medium. The authors use two fluorescent markers spaced by a rigid molecule to probe the statistics of the sum of the intensity at two points separated by tens to hundreds of nm. In a synthetic speckle field inside a clear medium, they find the expected behavior, namely that at a short distance the total intensity follows purely exponential statistics while at a larger distance the statistics become that of the sum of two independent speckle intensities.

In a medium loaded with scattering particles, a drastic change of the statistics is found, for a large part due to polarization scrambling by the scattering particles. Numerical calculations based on multi-particle Mie theory reproduce the experimental results well.

The work presents a new method and new insights in field correlations inside scattering media, considerably improving the state of knowledge from the existing oversimplified theory.

We thank the reviewer for their overall positive feedback evaluation of our work.

There are a few comments that I would like the authors to respond to before publication.

- On p4, ... our approach can be used to retrieve both n and S ... The “both” is confusing as n and S are dependent. If this is indeed the case, it should be phrased as “ n , or equivalently, S ” or similar. One cannot retrieve two independent quantities from a single scalar-valued measurement.

We agree with the referee.. The approach could be used for either n or, equivalently S . The wording has been adjusted in the manuscript:

Changes to the manuscript:

page 4:

“The inset reports the monotonic dependence of S or, equivalently, n as a function of N for a Gatta-sted with $D=90$ nm, showing that our approach can be used to retrieve either value in a dielectric medium even using a single D value.”

- While the measurement method employed by the authors can be applied to a reasonably general range of media, the results are extremely dependent on the exact composition of the random medium, i.e., the results are very unlikely to carry over to different random media. Moreover, the method only measures the correlations restricted to one component (the watery gel).

We agree with the Reviewer that our method is limited to probing two-point spatial correlations of light only in one component of the scattering medium (typically, the host phase). Nonetheless, this limitation does not affect the analysis that we performed, as we took care of evaluating numerically the differences between two-point spatial correlations in both components and for the host medium only.

Regarding the first point, we also agree with the Reviewer that in all generality, spatial light correlations will be extremely dependent on the microscopic details of a nanostructured sample. We deem this to be the main strength of our proposed technique: spatial coherence of light in 3D media can be seen as a quintessential example of a non-universal property due to its inherent near-field nature (see e.g., refs [31,47]). We therefore expect that the shape, size and polarization content of 3D speckle fluctuations will in general be strongly dependent on the nano-scale details of the scattering environment, giving rise to correlations that could be not measured otherwise from any far-field observable.

- Is it in principle possible to perform the same method with freely diffusing molecules, e.g. in tissue? This would be a very interesting prospect indeed.

Our method can be applied to diffusing molecules only if the time needed to collect enough statistical data is smaller than typical diffusion time. For our proof of concept, data on a single molecule can be collected in roughly 500 seconds (2500 frames at 0.2 sec exposure time). For this reason, an agarose gel solution was used to ensure that the nanorulers were not diffusing over such time scales. An improvement could be foreseen by substituting fluorescent molecules with quantum dots or analogue emitters less prone to bleaching. In this case, faster dynamics could be accessed by increasing illumination intensity to reduce exposure time.

Changes to the manuscript:

page 8:

“compared to a fluorescent molecule, a solid state emitter could also withstand higher excitation intensities, allowing faster measurement times and, therefore, the possibility to study the dynamics of speckle field correlations in slowly diffusing media.”

- What is the shape of the ZnO nanoparticles? From the manufacturer's information, I do not get the impression that they should be spherical. Even for non-spherical particles, one can expect reasonable agreement with Mie theory in the far field [M. I. Mishchenko, “Light

scattering by size-shape distributions of randomly oriented axially symmetric particles of a size comparable to a wavelength,” Appl. Opt. 32, 4652–4666(1993).], does this reasoning carry over to near field effects?

Thanks for raising this point. The answer to the first question can be found in ref. [42] in which a full characterization of the same particles is reported. Therein, the authors report results of XRD and SSA-BET measurements, showing that some particles may have a slightly ellipsoidal shape with deviations up to 8 nm from the spherical radius, and a comparable size polydispersity.

The second question requires instead a more articulated answer. This is an interesting open point which, up to now, could not be answered neither experimentally nor numerically, due to the physical inaccessibility of 3D media and the computational burden associated with their simulation, especially if comprising complex-shaped elements. On the experimental side, our method allows to lift the physical access limitation providing the first evidence that, at least for the subwavelength particle sizes analyzed here, small deviations from sphericity have a limited effect and the results agree very well (no free parameters) with numerical calculations for a collection of spheres with comparable size and polydispersity. The observed agreement could be partly explained by the fact that our fluorescent probes comprise several molecules which therefore provide an averaged response relative to a nm-scale volume. It is likely that any sphericity deviations below this scale are therefore averaged away.

However, it is worth stressing that, when studying samples with larger, arbitrary particle morphologies or altogether different nanostructured topologies, the technique we propose would be the only available method to probe and measure the resulting deviations, providing ground truth data against which future theories and numerical methods will be tested.

Changes to the manuscript:

The following paragraphs were added to the main text (page 6):

“It is worth noting that, due to the significant near-field contribution determining all previously described effects, samples comprising particles with different shape, size or arrangement may exhibit different types of deviations. In this respect, performing an ICSE characterization of nanostructured materials may allow them to vary their microscopic details in order to induce tailored constraints on the extent of internal intensity correlations and their average polarization content.”

- The depictions in the insets of Fig 3a,b seem a bit misleading as the particles should be much larger?

Absolutely – thanks for noticing this misleading depiction. We have now updated the figure with larger particles and added a note to the caption to clarify that the drawing is not a scaled representation of the particle and molecule sizes.

- Could extra information be extracted using time-correlated single-photon counting

As TCSPC allows to measure the local density of states, the idea to merge TCSPC and ICSE could be interesting to implement in order to study bulk speckle correlations inside samples with extreme scattering strengths ($k\ell < 10$), which are known to exhibit significant spatial fluctuations for the local density of states (see, e.g., [18]). In our case, however, the observed scattering mean free path of $\sim 8 \mu\text{m}$, is still in a scattering regime where $k\ell > 100$

and the local density of states is basically constant everywhere inside the sample, and determined solely by its effective refractive index.

Combining ICSE and TCSPC would allow to correct the observed fluorescence intensity by the local density of states. However, this would ideally require to adopt a collection scheme capable of resolving spatially the contributions from the two fluorescent markers, or a more elaborated analysis allowing to determine the two decay lifetimes separately.

and/or multiple excitation wavelengths?

Multiple-wavelength excitation could be in principle employed to study peculiar materials showing resonant behaviour such as photonic glasses made of monodispersed Mie particles or metamaterials supporting nanocavities. In that case one could probe different scattering regimes just by slightly detuning the optical excitation in and out of the resonant region.

On the other hand, the use of nanorulers decorated with mixtures of different emitters (i.e. emitters with excitation/emission at different wavelength such as Atto488 and Atto542) could be helpful to demultiplex the effect of local density of the states at different emission frequencies.

In light of the importance and originality of the manuscript, I advise publication of the manuscript after the authors have carefully considered the above remarks.

We thank again the referee for the careful referral and we believe we responded to all the points raised.

REVIEWERS' COMMENTS

Reviewer #1 (Remarks to the Author):

The authors have addressed my previous concerns in detail and also refuted some of my criticism adequately. They have also made changes to the manuscript and expressed their claims more carefully. The document has been improved substantially that way.

The article is very well written, and the study beautifully designed and novel. My concerns about the possible influence of the particles themselves, locally and via near field effects, on the short-range intensity correlation is now addressed on the bottom page 6. I can recommend publication of this work if the authors add a reference to chemical composition to their new sentence at the bottom of page 6:

"It is worth noting that, due to the significant near-field contribution determining all previously described effects, samples comprising particles with different shape, size, 'chemistry', or arrangement may exhibit different types of deviations."

Reviewer #2 (Remarks to the Author):

Here I provide a rather brief report to prevent further delay.

I found the authors responded to the previous reviewer comments in a careful and satisfactory manner, and advise to publish.

I have only very small comments: please check ref [38] and the punctuation around Eq.1

RESPONSE TO REVIEWERS' COMMENTS

Reviewer #1 (Remarks to the Author):

The authors have addressed my previous concerns in detail and also refuted some of my criticism adequately. They have also made changes to the manuscript and expressed their claims more carefully. The document has been improved substantially that way.

The article is very well written, and the study beautifully designed and novel. My concerns about the possible influence of the particles themselves, locally and via near field effects, on the short-range intensity correlation is now addressed on the bottom page 6. I can recommend publication of this work if the authors add a reference to chemical composition to their new sentence at the bottom of page 6:

"It is worth noting that, due to the significant near-field contribution determining all previously described effects, samples comprising particles with different shape, size, 'chemistry', or arrangement may exhibit different types of deviations."

Reviewer #1

We are glad for the very positive feedback.

The sentence at the bottom of page 6 has been modified as suggested by the Reviewer.

Reviewer #2 (Remarks to the Author):

Here I provide a rather brief report to prevent further delay.

I found the authors responded to the previous reviewer comments in a careful and satisfactory manner, and advise to publish.

I have only very small comments: please check ref [38] and the punctuation around Eq. 1

Reviewer #2

We thank the Reviewer for noticing the misprint at ref. [38] and the missing punctuation around Eq. 1, which we have fixed.